# Appropriateness of Dyslipidemia Management Strategies in Post-Acute Coronary Syndrome: A 2023 Update

**DOI:** 10.3390/metabo13080916

**Published:** 2023-08-04

**Authors:** Fabiana Lucà, Fabrizio Oliva, Carmelo Massimiliano Rao, Maurizio Giuseppe Abrignani, Antonio Francesco Amico, Stefania Angela Di Fusco, Giorgio Caretta, Irene Di Matteo, Concetta Di Nora, Anna Pilleri, Roberto Ceravolo, Roberta Rossini, Carmine Riccio, Massimo Grimaldi, Furio Colivicchi, Michele Massimo Gulizia

**Affiliations:** 1Cardiology Department, Grande Ospedale Metropolitano, AO Bianchi Melacrino Morelli, 89129 Reggio Calabria, Italy; fabiana.luca92@gmail.com; 2De Gasperis Cardio Center, Niguarda Hospital, 20162 Milan, Italy; 3Operative Unit of Cardiology, P. Borsellino Hospital, Marsala, ASP Trapani, 91025 Trapani, Italy; 4CCU-Cardiology Unit, Ospedale San Giuseppe da Copertino Hospital, 73043 Copertino, Italy; 5Clinical and Rehabilitation Cardiology Department, San Filippo Neri Hospital, ASL Roma 1, 00100 Roma, Italy; 6Sant’Andrea Hospital, ASL 5 Regione Liguria, 19124 La Spezia, Italy; 7Department of Cardiothoracic Science, Azienda Sanitaria Universitaria Integrata di Udine, 33100 Udine, Italy; 8Cardiology Unit, Brotzu Hospital, 09121 Cagliari, Italy; 9Cardiology Department, Giovanni Paolo II Hospital, 88046 Lamezia Terme, Italy; 10Cardiology Unit, Ospedale Santa Croce e Carle, 12100 Cuneo, Italy; 11Cardiovascular Department, Sant’Anna e San Sebastiano Hospital, 81100 Caserta, Italy; 12Department of Cardiology, General Regional Hospital “F. Miulli”, 70021 Bari, Italy; 13Cardiology Department, Garibaldi Nesima Hospital, 95122 Catania, Italy

**Keywords:** lipid-lowering therapy (LLT), post-acute-coronary-syndrome, PCSK9 inhibitors, inclisiran, statins, ezetimibe, bempedoic acid, adherence

## Abstract

It has been consistently demonstrated that circulating lipids and particularly low-density lipoprotein cholesterol (LDL-C) play a significant role in the development of coronary artery disease (CAD). Several trials have been focused on the reduction of LDL-C values in order to interfere with atherothrombotic progression. Importantly, for patients who experience acute coronary syndrome (ACS), there is a 20% likelihood of cardiovascular (CV) event recurrence within the two years following the index event. Moreover, the mortality within five years remains considerable, ranging between 19 and 22%. According to the latest guidelines, one of the main goals to achieve in ACS is an early improvement of the lipid profile. The evidence-based lipid pharmacological strategy after ACS has recently been enhanced. Although novel lipid-lowering drugs have different targets, the result is always the overexpression of LDL receptors (LDL-R), increased uptake of LDL-C, and lower LDL-C plasmatic levels. Statins, ezetimibe, and PCSK9 inhibitors have been shown to be safe and effective in the post-ACS setting, providing a consistent decrease in ischemic event recurrence. However, these drugs remain largely underprescribed, and the consistent discrepancy between real-world data and guideline recommendations in terms of achieved LDL-C levels represents a leading issue in secondary prevention. Although the cost-effectiveness of these new therapeutic advancements has been clearly demonstrated, many concerns about the cost of some newer agents continue to limit their use, affecting the outcome of patients who experienced ACS. In spite of the fact that according to the current recommendations, a stepwise lipid-lowering approach should be adopted, several more recent data suggest a "strike early and strike strong" strategy, based on the immediate use of statins and, eventually, a dual lipid-lowering therapy, reducing as much as possible the changes in lipid-lowering drugs after ACS. This review aims to discuss the possible lipid-lowering strategies in post-ACS and to identify those patients who might benefit most from more powerful treatments and up-to-date management.

## 1. Introduction

Coronary artery disease (CAD) is still a leading cause of global mortality, in spite of interventional and pharmacological strategy improvements [1].

Acute coronary syndrome (ACS) represents one of the most severe clinical presentations of coronary artery disease (CAD) [2].

In the last decades, ACS management has been considerably improved, leading to a significant reduction in in-hospital mortality, from 30% to 3–8% [3].

Remarkably, at five years, the mortality rate has been estimated to be 19–22% [4,5].

Moreover, in the 24 months following an ACS event, the survivors have a 30% likelihood of experiencing a second event [6].

The recurrence of ischemic cardiovascular (CV) events and, consequently, the need for new revascularizations has been shown to be associated with higher mortality in the long run [7,8,9,10].

Nowadays, it has been well-recognized that dyslipidemia is the main cause of atherosclerosis development. Therefore, it should be adequately treated in order to achieve a significant reduction in CV risk, especially in patients who experienced ACS. Indeed, a lower rate of CV events and mortality has been shown to be associated with effective lipid management in this group of patients [5].

In the latest years, lipid-lowering strategies have been noticeably improved. Because of this, physicians may have several options to treat this subset of patients.

An early lipid-lowering pharmacological approach at hospital discharge combined with short-term follow-up has been recently proposed in order to reduce adverse events in post-ACS patients [11].

Nonetheless, post-ACS management remains suboptimal, particularly in terms of lipid therapeutic target achievement.

This review aims to promote an understanding of the main therapeutic options, with a special focus on those recently introduced in clinical practice, for lipid management in patients who experienced ACS.

## 2. Lipid-Lowering Therapy (LLT)

Dyslipidemia is a metabolic disorder determined by the concurrence of genetic conditions and unhealthy lifestyles [12].

A close relationship between the incidence of atherosclerosis and serum cholesterol levels has been well recognized [13], and increased values of low-density lipoprotein cholesterol (LDL-C) are the primary cause of the development and progression of atherosclerosis [14].

Indeed inflammation, LDL-C, platelet activation, and endothelial dysfunction have been considered the leading atherogenic factors [15,16]. Remarkably, it has been shown that LDL-C and circulating monocyte levels are linked, confirming the correlation between lipids, inflammatory status, and CAD progression [17,18].

Furthermore, it has been established that intensive lipid-lowering therapy (LLT) may improve plaque phenotype, contributing to plaque stabilization [19,20].

Moreover, it has been claimed that an intensive LLT is correlated with better outcomes in those patients who experienced ACS [21,22].

Consequently, the reduction [21,23] of circulating LDL-C is one of the most relevant goals to achieve for CVD prevention. This goal is achievable thanks to several effective pharmacological interventions currently available [24]. Table 1 summarizes the action of LLT.

### 2.1. Statins

Statin therapy has been shown to decrease all-cause mortality and 5-year incidence of major adverse cardiovascular events (MACE) by 12% and 21%, respectively, per mmol/L LDL-C reduced (roughly equivalent to 39 mg/dL) [37]. A 20% reduction of CV adverse events rate has been reported using statins compared with placebo and high-intensity statins compared with low-intensity statins for each LDL 1.0-mmol/L reduction [23].

Nowadays, statins are considered the first-line pharmacological therapy in order to manage dyslipidemia and reduce CV risk [38]. Some statins derive from fungal fermentation, such as lovastatin, pravastatin, and simvastatin [39], others from synthetic processes (atorvastatin, rosuvastatin) [39].

It has been shown that statins competitively inhibit the activity of 3-hydroxy-3-methyl-glutaryl-coenzyme A reductase (HMGR), which converts 3-hydroxy-3-methyl-glutaryl-coenzyme A (HMG-CoA) into mevalonic acid, a cholesterol precursor [40].

This phase is an early rate-limiting step in cholesterol biosynthesis. The binding of statins with HMG-CoA reductase is reversible [41].

As a result of statin activity, a non-linear dose-dependent LDL-C reduction occurs.

Considering the fact that mevalonate, derived from HMGR, is also the precursor of many other nonsteroidal isoprenoid compounds, such as farnesyl pyrophosphate (FPP) and geranylgeranyl pyrophosphate (FPD), statins also affect the Ras-related small GTPase signaling pathway (Ras and Rho) [42].

Some of the statins’ pleiotropic effects are ascribed to the inhibition of these intracellular isoprenoid-dependent proteins [43]. Indeed, several cardioprotective effects of statins observed during chronic use have been thought to be not directly linked to cholesterol levels [44,45,46].

Anti-inflammatory activity has also been postulated [47]. A potent modulating effect on endothelial cell nitric oxide synthase (eNOS) resulting in the upregulation of eNOS enzyme and a decrease in nitric oxide (NO) production [47], as well as a reduction in cytokine C-reactive protein (CRP) levels, has been reported [31,48,49,50].

A large number of experimental and clinical studies investigated the potential additional effects of statins, postulating an improvement in endothelial function and vascular tone, plaque stabilization effects and anti-thrombotic activity, and reduction in oxidative stress [47].

An incremental lowering of LDL-C values, which has been shown in patients receiving intensive statin therapy compared with those treated with moderate-dose statins, results in a lower rate of nonfatal CV events [21,30,51,52,53]. Good tolerance has been generally reported in patients treated with statins, but 20% of intolerant patients reported statin intolerance syndrome with adverse effects on muscles, varying from myalgia to myopathy, myositis, and rhabdomyolysis [54,55]. Statin-induced intolerance may cause therapy interruption [56,57].

A rise in the risk of adverse CV outcomes has been reported in patients discontinuing statin therapy [58,59]. A genetic predisposition has been hypothesized to be involved in the development of statin-induced muscle failure [60].

However, safety issues associated with intensive statin therapy and the evidence of residual risk of recurrent CV events [61] have led to the introduction of additional non-statin therapies in clinical practice [62].

### 2.2. Ezetimibe

Ezetimibe joins a new drug class of selective cholesterol absorption inhibitors that block the internalization of cholesterol into enterocytes at the level of the brush border of the small intestine [63].

The ezetimibe-mediated inhibition of the Niemann–Pick C1-like 1 (NPC1L1) polytopic transmembrane protein results in reduced intestinal cholesterol absorption [64].

A 10–14% and 23–24% LDL-C plasma level reduction has been observed in patients treated with ezetimibe alone or in addition to statins, respectively [65,66]. Ezetimibe combined with a low dose of statins may represent a suitable option in case of symptoms of intolerance in patients treated with full doses of statins [65]. More recent studies have shown great results with ezetimibe and bempedoic acid co-therapy, with a 38% mean difference in LDL cholesterol level reduction compared to the placebo [67].

In the IMPROVE-IT trial, in high-risk patients post ACS, the combination strategy of ezetimibe 10 mg and simvastatin 40 mg proved to be superior to simvastatin 40 mg alone in lowering the recurrence of CV events, irrespective of baseline LDL-C levels [68]. An incremental beneficial effect of ezetimibe added to statin has been observed in patients with DM and in those without DM but at high risk of recurrent CV events [69].

### 2.3. Bempedoic Acid

Bempedoic acid has recently entered the pharmacological armamentarium for dyslipidemia treatment [70].

After its conversion to the active metabolite by acyl-CoA synthetase 1 (ACSVL1), exclusively expressed in liver cells, bempedoic acid lowers cholesterol synthesis by inhibiting adenosine triphosphate (ATP) citrate lyase, which, in the enzymatic cascade that leads to cholesterol synthesis, acts upstream of HMGCR.

Similarly to statins, reduced hepatic cholesterol synthesis induced by bempedoic acid leads to the upregulation of LDL-R expression and, consequently, reduction in LDL-C levels [71]. The reason why fewer muscular adverse effects have been associated with this therapy is that bempedoic acid is a prodrug selectively activated in the hepatic tissue. In skeletal muscle, the prodrug can not be activated due to the absence of ACSVL1, explaining the reduction in adverse muscle effects mentioned above. Moreover, ATP citrate lyase downregulation and AMP-activated protein kinase (AMPK) upregulation improves glucose metabolism regulation [60] and reduces the inflammatory pathway and cytokine production [72]

The safety and efficacy of the long-term use of bempedoic acid have been investigated in several clinical trials, including Cholesterol Lowering via BEmpedoic Acid, an ACL-inhibiting Regimen (CLEAR) Tranquility [67], CLEAR Serenity [73], CLEAR Wisdom [73], and CLEAR Harmony [74,75]. At a daily dose of 180 mg, an LDL-C reduction from 17.4 to 28.5% was obtained [76].

Recently, in a trial that included 13,970 patients, 69.9% with a previous CV event with statin intolerance, the incidence of primary endpoint events (death from CV causes, nonfatal myocardial infarction, nonfatal stroke, or coronary revascularization) was 13% lower in the treated group. The incidences of gout and cholelithiasis were higher with bempedoic acid than with placebo (3.1% vs. 2.1% and 2.2% vs. 1.2%, respectively), as were the incidences of small increases in serum creatinine, uric acid, and hepatic enzyme levels [77].

### 2.4. PCSK9 Inhibitors

Proprotein convertase subtilisin-like kexin type 9 (PCSK9) is a serine protease mainly expressed in the liver that targets LDL-Rs, promoting their lysosomal degradation and decreasing circulating LDL-C clearance [78]. PCSK9 monoclonal antibodies (mAbs) selectively bind to extracellular PCSK9, preventing LDL-R degradation and lowering plasma LDL-C levels. Two fully human mAbs, Alirocumab and Evolocumab, have been approved by FDA and EMA [79].

Statin treatment increases circulating PCSK9 serum levels; consequently, the greatest effect of these mAbs has been observed when used in combination with statins [80]. A reduction in LDL-C plasma levels has been shown, of up to 65% for alirocumab and 80% for evolocumab, following an injection every 2 or 4 weeks [81].

PCSK9 mAbs were associated with a 20% lower risk of myocardial infarction, a 22% lower risk of ischemic stroke, and a 17% lower risk of coronary revascularization [82]. Their use was associated with a favorable safety profile without increasing risk of neurocognitive adverse events, liver enzyme elevations, rhabdomyolysis, or new-onset diabetes mellitus. According to the GLAGOV data [83], both molecules have been shown to favor morphological stabilization and reduction of carotid plaques [20,84,85,86,87], delaying ASCVD progression.

### 2.5. Inclisiran

Small interfering RNA (siRNA) molecules now represent the next generation of drugs designed to antagonize PCSK9. Inclisiran is an siRNA specific for PCSK9 that prevents the translation of PCSK9 messenger RNA, leading to decreased concentrations of the protein and lower concentrations of LDL cholesterol.

Inclisiran blocks the expression of a specific gene by selectively silencing the translation of PCSK9 messenger RNA (mRNA) [88], leading to a long-lasting reduction in LDL-C even up to 12 months [89,90]. It was thought that the reason why inclisiran has such long-term efficacy is that the silencing complex remained active even after mRNA degradation, resulting in a considerable and long-lasting reduction in plasma LDL-C levels [89]. Consequently, inclisiran has been considered an attractive therapeutic option, particularly for non-adherent patients [91]. What the impact of inclisiran on reducing lipoproteins and MACE is, has been largely investigated in the ORION/VICTORION studies [90,92,93,94,95,96,97,98,99,100,101,102,103,104,105,106], which evidenced a decrease in LDL-C over 1 year of 29.5–38.7% and 29.9–46.4% after a single dose and after two doses, respectively (*p* < 0.001). Moreover, Lp(a) has been shown to significantly decrease. 

## 3. Historical Randomized Controlled Trials with Statin and Ezetimibe

Statins are the first drugs with a marked and sustained effect on the reduction of LDL-C that have been extensively studied in several clinical trials. Statin trials have played a pivotal role in demonstrating the effects of lipid-lowering therapies (LLT) in reducing CV risk in both primary and secondary prevention. These trials have consistently shown that statin therapy reduces the risk of major CV events by approximately 20–50% among different populations at high CV risk. Different trials focused on the early initiation of statin therapy in patients with ACS [107]. Indeed, starting treatment early after ACS allows for the potential benefits of statins to be maximized. During this acute phase, aggressive lipid-lowering therapy can have a substantial impact on reducing plaque instability, inflammation, and subsequent CV events. The Pravastatin in Acute Coronary Treatment (PACT) trial and the Fluvastatin On Risk Diminishing After Acute myocardial infarction (FLORIDA) trial both showed that moderate-intensity statin treatment did not significantly reduce the early incidence of major CV events [29,108]. The Myocardial Ischemia Reduction with Aggressive Cholesterol Lowering (MIRACL) trial demonstrated that early initiation of high-dose atorvastatin in patients with unstable angina or non-ST-segment elevation myocardial infarction resulted in a reduction in MACE. Patients in the atorvastatin group had a 16% reduction in the risk of the primary endpoint, which included death from any cause, non-fatal myocardial infarction, cardiac arrest, or recurrent symptomatic myocardial ischemia requiring emergency rehospitalization [31]. The Aggrastat to Zocor (A to Z) trial compared early intensive statin therapy with simvastatin followed by a switch to high-dose atorvastatin versus standard-dose simvastatin in patients with ACS [30]. The trial found that the early intensive therapy group had a lower incidence of MACE compared to the standard therapy group. At 30 days, the early intensive therapy group had a 16% reduction in the composite endpoint of death from any cause, non-fatal myocardial infarction, readmission for ACS, or stroke. However, there was no significant difference between the two groups in the primary endpoint of death from CV causes, non-fatal myocardial infarction, or resuscitated cardiac arrest. The Pravastatin or Atorvastatin Evaluation and Infection Therapy-Thrombolysis In Myocardial Infarction 22 (PROVE-IT TIMI 22) trial compared the effectiveness of high-dose atorvastatin versus standard-dose pravastatin in reducing CV events among patients with recent ACS [21]. The event rate was 26.3% in the atorvastatin group and 32.7% in the pravastatin group, representing a relative risk reduction of 16% in favor of high-dose atorvastatin. These data provided evidence that high-intensity statin treatment after ACS improves patient outcomes and reduces the burden of CV disorders. The rationale for the benefit of a more intensive lipid-lowering therapy has been reinforced by the Treating to New Targets (TNT) study, which demonstrated that intensive lipid-lowering therapy with atorvastatin 80 mg resulted in a significant reduction in MACE compared to moderate therapy with atorvastatin 10 mg in patients with stable CAD [51].

In addition to statins, the IMPROVE-IT trial investigated the role of ezetimibe, a cholesterol absorption inhibitor, in further reducing CV risk [36]. The trial enrolled over 18,000 patients with recent ACS and showed that adding ezetimibe led to a 6.4% reduction in the composite endpoint of CV death, major coronary event, or non-fatal stroke compared to statin therapy alone. Despite the relatively modest effect on the outcome, the IMPROVE-IT trial provided important evidence supporting the LDL-C hypothesis and highlighted the incremental benefit of combining other therapies with statins in reducing MACE in patients with recent ACS (Table 2).

## 4. Evidence on PCSK9 Inhibitors in Post-ACS

In post-acute ACS, and more generally in patients with high and very high CV risk, LDL-C reduction is the foundation of secondary prevention management. Randomized clinical trials have demonstrated that, after an ACS event, early treatment with a high-efficacy statin, such as atorvastatin 80 mg, leads to an early clinical benefit in terms of MACE reduction. Due to the non-negligible rate of ACS patients who do not achieve LDL-C level targets with the use of statin therapy alone, further therapeutic options should be implemented. PCSK9 has been found to be a therapeutic target to reduce LDL-C levels effectively and powerfully. Over the last decades, basic and clinical research have led to the development of two monoclonal antibodies, alirocumab and evolocumab, which are able to lower LDL-C levels and improve prognosis. In addition to high-intensity statin treatment, PCSK9 antibodies have been shown to reduce LDL-C by up to 75% compared to placebo. In the randomized controlled trial ODYSSEY outcome, alirocumab was tested in patients treated with high-intensity statins and a recent ACS (1–12 months) [109]. Patients treated with alirocumab had a lower incidence of the primary endpoint, which was a composite of MACE, including death from CAD, nonfatal acute myocardial infarction (AMI), ischemic stroke, or unstable angina requiring hospitalization. In addition, alirocumab, on top of optimal lipid-lowering treatment, was associated with reduced mortality among patients with more than a three-year follow-up [90]. In this study, patients with greater LDL-C levels at baseline (>100 mg/dL) had the greater benefit in terms of mortality reduction. Alirocumab also reduced lipoprotein (a) levels by 5.0 mg/dL, and Lp(a) reduction predicted lower MACE incidence [110].

In a prespecified analysis of the Further Cardiovascular Outcomes Research With PCSK9 Inhibition in Subjects With Elevated Risk (FOURIER) trial that examined data from 5711 patients with a recent myocardial infarction (<12 months), evolocumab reduced the incidence of the primary endpoint (a composite of CV death, AMI, stroke, coronary revascularization, or hospitalization for unstable angina) and key secondary endpoint (a composite of CV death, AMI, or stroke) by 19% and 25%, respectively [111]. Due to the higher incidence of events among patients with a recent myocardial infarction, in these patients, the event risk reduction at three-year follow-up was greater compared to those who had a myocardial infarction > 12 months before randomization. These results confirm that earlier achievement of lower LDL-C levels confers a greater clinical benefit. The Evolocumab for Early Reduction of LDL Cholesterol Levels in Patients With Acute Coronary Syndromes (EVOPACS) study investigated lipid-lowering treatment with evolocumab 420 mg started during hospitalization for ACS. In this study, patients were randomized to receive evolocumab or placebo in addition to atorvastatin 40 mg. Evolocumab was administered during the index hospitalization, but the interval times from ACS symptoms’ presentation and the first dose of drug administration were not reported. Mean LDL-C levels at enrollment were 136 ± 38 mg/dL. Evolocumab was found to be well tolerated and to lead to a significant LDL-C reduction (mean percentage changes vs. placebo −40.7%; 95% confidence interval (CI) −45.2 to −36.2%; *p* < 0.001) with >95% of treated patients that achieved recommended LDL-C target levels at week 8 after ACS [33]. The Evolocumab in Acute Coronary Syndrome (EVACS) study [112] is a randomized controlled trial that tested evolocumab 420 mg vs. placebo, on top of high-intensity statins, in patients with non-ST-segment elevation myocardial infarction ( NSTEMI). In this study, evolocumab was administered within 24 hours of presentation. At hospital discharge, the percentage of patients treated with evolocumab that achieved LDL-C levels recommended by 2019 European Society of Cardiology guidelines (<55 mg/dL) was higher than that of patients in the placebo group (65.4% vs. 23.8%, respectively; *p* < 0.01).

The randomized, sham-controlled, double-blind EPIC-STEMI trial evaluated the change in LDL-C when alirocumab was used in patients with ST-segment elevation myocardial infarction (STEMI) treated with primary percutaneous coronary intervention (PCI). Alirocumab 150 mg administered before primary PCI, at 2 weeks and at 4 weeks led to a 72.9% reduction in LDL-C level at 6 weeks as compared to a 48.1% reduction in the sham-control group (*p* < 0.001) [34].

A further small placebo-controlled, double-blind study randomized 20 patients 1:1 to receive alirocumab 150 mg or placebo within 24 hours of NSTEMI presentation. At 14 days from treatment administration, in the group managed with alirocumab, a significant reduction in LDL-C level (−64 mg/dL, −96 to −47 mg/dL) was found as compared with the placebo group (+1 mg/dL, −25 to +16 mg/dL) [35].

Overall, available evidence on the use of anti-PCSK9 monoclonal antibodies in the ACS setting consistently shows a powerful and early effect of these drugs in terms of LDL-C reduction and better prognosis (Table 3), without relevant safety issues. These findings support the use of anti-PCSK9 monoclonal antibodies in the early phase of ACS.

In more recent years, pharmacological research has led to the development of PCSK9 inhibitors that act by interfering with PCSK9 hepatic synthesis. Inclisiran, a small interfering RNA agent, is a drug in a more advanced phase of development. When tested in patients with ASCVD, inclisiran has been found to reduce LDL-C levels by up to 53% [102]. However, no evidence on the specific setting of ACS or on clinical outcomes is currently available.

The VICTORION-INCEPTION trial is currently ongoing. This is a phase IIIb, randomized, open-label study aimed at assessing inclisiran effectiveness when administered on top of usual care in patients within 5 weeks from an ACS and with LDL-C levels ≥ 70 mg/dL, despite statin treatment [113].

## 5. Current Guideline Recommendations

Patients who present with ACS are at increased risk of experiencing recurrent CV events, especially within the first year after hospital discharge. LDL-C has been shown to be a causal factor for the development of atherosclerosis, and a strong relationship between pharmacologic LDL-C lowering and a reduction in CV events post-ACS has been shown. According to the ESC/EAS guidelines [25], it has been recommended to initiate a high-intensity statin up to the highest tolerated dose in all statin-naive ACS patients with no contraindication; a 50% LDL-C reduction from baseline value and an LDL-C goal of <1.4 mmol/L (<55 mg/dL) is recommended. In those with recurrent events within 2 years, a goal of <1.0 mmol/L (<40 mg/dL) for LDL-C should be considered.

LDL-C levels tend to decrease during the first 24 h after ACS, and the lower value can be measured 7 days after the event (a mean reduction in LDL-C of 10% approximately); therefore, a lipid profile should be assessed as soon as possible after admission for ACS. LLT should be initiated as soon as possible. The latest ESC/EAS guidelines [25] still recommend a stepwise approach. Lipid levels should be re-evaluated 4–6 weeks after ACS, and if the LDL-C goal is not achieved, a combination with ezetimibe is recommended.

Statins are generally well tolerated; indeed, myopathy and muscle symptoms are not frequently reported. Among patients in whom statins cannot be prescribed due to intolerance or adverse effects, it has been suggested to start with PCSK9 inhibitors in combination with ezetimibe. PCSK9 inhibitors also lower triglycerides, raise HDL-C and apolipoprotein A-I, and lower lipoprotein(a). In patients presenting with ACS already on a high-intensity statin and/or ezetimibe, it should be advisable to prescribe PCSK9 inhibitors before discharge.

The 2018 AHA/ACC Guidelines recommended intensifying the approach to LDL-C reduction in secondary prevention even though their target is different from ours (≤70 mg/dL) [114].

The following ESC guidelines on CVD prevention and NSTEMI [26,115] have not changed the treatment target for LDL-C, confirming the step-by-step approach, also in secondary prevention (Table 4).

## 6. Real-World Data

Many observational studies have been designed to examine the level of hypolipidemic drug treatment in patients with CAD, namely post-acute coronary syndromes (ACS) in the real world, as well as to provide contemporary data on the implementation of guideline recommendations for LLT across different settings and populations, e.g., how this impacts LDL-C goal achievement.

The main studies assessing these items were the European Action on Secondary Prevention through Intervention to Reduce Events (EUROASPIRE) I, II, III, IV, and V studies [116,117,118,119,120], the EU-Wide Cross-Sectional Observational Study of Lipid-Modifying Therapy Use in Secondary and Primary Care (DA VINCI) [121], the DYSIS-China [122], the DYSIS II Study [123], the TARGET study [124], The Hyperlipidaemia Therapy in tERtiary Cardiological cEnTer (TERCET) Registry [125], The Acute Coronary Syndrome Management (ACOSYM) registry [126], the ACS patient pathway project [127], and the HYDRA-ACS registry [128].

In the first EUROASPIRE, only 33% of the patients received LLT [116]. The use of LLT was relatively higher (60.9%) in EUROASPIRE II [117]; however, the most frequently used doses of LLT agents were much lower than the doses of proven effect used in clinical trials [117]. The use of statins was 78% in the EUROASPIRE III [118], with wide variations between countries.

Of patients at high and very high CV risk from the Cegedim Longitudinal Practice Database in Germany from 2011 to 2013, only 35% received current statin treatment [129]. In the Greek TARGET study, at the end of the follow-up, 87.7% of patients remained on statin treatment [124].

EUROASPIRE V [130] highlighted that the percentage of CAD adults receiving statins at ìLDL goal in ACS patients has not been achieved [27,36,131].

There are several factors that impact LDL-C goal achievement after ACS discharge. By and large, they may be differentiated as those related to the patient, those related to physicians, and those related to the socio-sanitary system (Figure 1). High-intensity statins combined with ezetimibe have been shown to improve adherence, so this combination should be preferred at discharge [132,133]. The monthly and yearly administration of PCSK9 antibodies and siRNA, respectively, have been related to improved adherence and prognosis.

Nonadherence has been estimated at 59.2% after 1 year [134]. Medication adherence largely influences life-long risk [135], and an increase in CV risk of 5% has been described for 10% of LLT nonadherence [136].

LDL-C control seems better in men than in women [120,123,137,138]. Baseline LDL-C level is obviously negatively associated with goal achievement rate [137]. Regarding common comorbidities, patients with obesity [123], concomitant coronary heart disease (CHD), and peripheral arterial disease (PAD) or diabetes were least likely to reach their target LDL-C goals [139]. However, discordant data about diabetic patients and those with chronic kidney disease (CKD) have been reported [120,123,138]. Better adherence is obviously associated with an improved goal achievement rate [137]. Additionally, patients who quit smoking have a better LDL-goal achievement rate than smokers [123,137], and more patients with recent ACS achieved the LDL-C target (33.8%) compared to chronic CHD patients (21.2%) [122]; this may be a proxy of self-empowerment in facing the postinfarction period. Statin side effects (real or perceived) and intolerance, with consequent treatment withdrawn, may obviously also play a role in the gap between LDL goal and LDL value achieved in clinical practice.

Clinical inertia by the physician (both in starting and reinforcing treatments) appears to be a fundamental factor, too. LDL-C control, in fact, is better in those on a high-intensity LLT compared to those on low- or moderate-intensity LLT [120,123,126,140,141], as well as in those prescribed statin and ezetimibe combinations than in those prescribed intensive statin monotherapy [137,142]. Goal attainment for secondary prevention was 45% for patients on statins only, 53% for those receiving combination therapy with ezetimibe, and 67% for those receiving combination therapy with PCSK9 inhibitors, respectively [121]. Stating a target in discharge documentation was associated with significant improvements in lipid testing and patients achieving LDL-C targets [143].

The role of the sociosanitary system is clear, not only due to variations in protocols and practices across health systems, e.g., with regulatory obstacles (consequent to the high cost of new-generation agents), but also in the availability of cardio-rehabilitation. Patients that have participated in cardiac rehabilitation programs after a CV event, in fact, show the best levels of LDL-C target achievement [144]. In addition, different reimbursement politics may partly explain some geographical variations in undertreatment.

## 7. Future Perspectives

As previously emphasized, there is a gap between guideline recommendations and current clinical practice. In the real world, a low proportion of patients reach the recommended targets, underlying the need for new therapeutic strategies. Recent studies have demonstrated the safety and efficacy of an early and significant reduction in LDL-C levels in patients with ACS. In an ideal post-ACS scenario, we would need 12 weeks for patients to receive optimal LLT if the stepwise approach suggested by the guidelines is implemented. However, the first 100 days correspond to the most vulnerable phase after an ACS.

A fast-track strategy has been recently proposed for post-ACS patients [11] and those who experienced an AMI within the last 12 months or had multiple previous CV events [145].

Recently, a panel of Italian cardiologists on the use of a mini-Dolphy technique has reported a broad consensus on the early systematic use of combination therapies in post-ACS patients [146]

In-hospital initiation of PCSK9 inhibitors enables achieving LDL-C goals in the majority of patients in the most vulnerable phase.

Musumeci et al. [147] proposed a triple-therapy approach (high-intensity statins, ezetimibe, and PCSK9 inhibitors) immediately after discharge, in those patients who recently experienced an ACS and were considered to be at prohibitive risk due to their clinical, angiographic, and lipidic features, in order to achieve an early and significant LDL-C reduction. This strategy, according to previous data [135,145], is based on the risk profile of patients and takes into account several factors such as multiple CV events, multiple ASCVD localizations, and multivessel CAD. Additionally, a tailored and early approach recently proposed by De Luca et al. [11] suggests the intensive use of LLT and drug refund requirements, also highlighting the importance of short-term follow-up.

Notably, it has been suggested to take into account the expected reduction value for the single drug used.

According to this statement [11], statin-naive patients in the post-ACS period should start high-intensity statin therapy during hospitalization in order to obtain a 50% reduction in LDL-C, adding also ezetimibe [146,148].

Conversely, in those patients previously treated with statins, the type and dose of statin should be considered, and the use of a more-intensive LLT should be evaluated. For intolerant patients, ezetimibe and PCSK9 inhibitor use are advisable [11]

Notably, it is the patients with baseline LDL-C values higher than 100 mg/dL that are expected to benefit from an early combined LLT approach the most. This is due to the fact that these patients are unlikely to lower LDL by more than 55 mg/dL using only high-intensity statins; hence, in this context, a double or triple LLT approach should be reasonable. And now, the “strike early and strong approach” is considered to be the most advisable one [27].

## 8. Conclusions

In conclusion, gaps between clinical guidelines and clinical practice for lipid management persist across Europe. Too many CAD patients with dyslipidemia are still inadequately treated, and LDL-C control is suboptimal; the available LLT armamentarium, including combination therapy, remains largely underused in high- and very-high-risk patients, leading to suboptimal management of residual risk.

There is a strong need for novel strategies to intensify lipid-modifying management and improve long-term LDL-C control after ACS. More-effective strategies for the management of dyslipidemia are needed in order to minimize the discrepancy between real-world clinical practice and current guideline recommendations.

## Figures and Tables

**Figure 1 metabolites-13-00916-f001:**
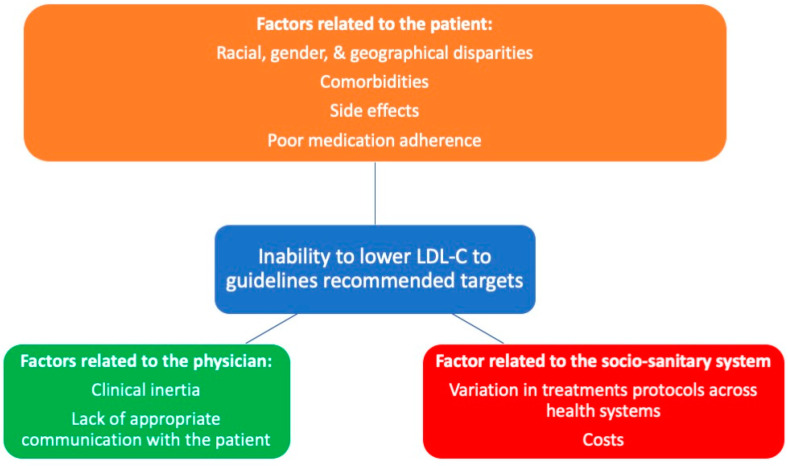
Factors affecting LDL-C goal achievement.

**Table 1 metabolites-13-00916-t001:** Main lipid-lowering drugs.

Drug Classes	Mechanism of Action	Expected Proportional LDL-C Reduction (vs. Placebo)	Main RCTs after ACS
Statins (Moderate Intensity): Atorvastatin 10–20 mg; Rosuvastatin 5–10 mg; Simvastatin 20–40 mg, etc.	Inhibit the activity of 3-hydroxy-3-methyl-glutaryl-coenzyme A reductase	30% [25,26,27]	FLORIDA [28], PACT [29], A to Z [30]
Statins (High Intensity): Atorvastatin 40–80 mg; Rosuvastatin 20–40 mg		50% [25,26,27]	MIRACL [31], PROVE-IT TIMI 22 [21]
Ezetimibe	Inhibits the Niemann–Pick C1-like 1 transmembrane protein	20% [27]	
Bempedoic Acid	Inhibits adenosine triphosphate citrate lyase	15–25% [27]	CLEAR ACS [32] (ongoing)
PCSK9-i (Alirocumab, Evolocumab)	Monoclonal antibodies which selectively bind to extracellular PCSK9, preventing LDL-R degradation	60% [25,26,27]	EVOPACS [33], EPIC-STEMI [34], VCU-AlirocRT [35]
PCSK9 siRNA (Inclisiran)	Prevent the translation of PCSK9 messenger RNA	50% [27]	VICTORION-INCEPTION (ongoing)
Statin + Ezetimibe	Combined	Maximum 65% [25,26,27]	IMPROVE-IT [36]
Bempedoic Acid + Ezetimibe	Combined	35% [27]	
High Intensity Statin + PCSK9-i	Combined	75% [25,26,27]	
High Intensity Statin + Ezetimibe + PCSK9-i	Combined	85% [25,26,27]	

Abbreviations: ACS: Acute coronary syndrome; LDL-C: Low-density lipoprotein cholesterol; LDL-R: Low-density lipoprotein receptors; PCSK9: Proprotein Convertase Subtilisin/Kexin Type 9 (PCSK9) inhibitors; RCTs: Randomized controlled trials; siRNA: Small Interfering ribonucleic acid.

**Table 2 metabolites-13-00916-t002:** Landmark statin trials in secondary prevention after Acute Coronary Syndromes.

Trial Name	Year of Publication	Participants	Statin Used	Endpoint Measured	HR (95% CI)	Key Findings
MIRACL [31]	2001	3086 patients with ACS	Atorvastatin, 80 mg, vs. placebo	Composite endpoint of death, myocardial infarction, or cardiac arrest	0.84 (0.70–1.00)	Trend toward reduction in composite endpoint
FLORIDA [28]	2002	540 patients with AMI	Fluvastatin 80 mg vs. placebo	Composite of major CV events or residual ischemia	-	No difference in primary endpoint
PACT [29]	2004	3408 with ACS	Pravastatin 20/40 mg vs. placebo	Composite of death, MI, revascularization, or stroke	0.94 (0.72–1.13)	Pravastatin did not significantly reduce major CV events compared to placebo in ACS patients.
PROVE IT-TIMI 22 [21]	2004	4162 patients with ACS	Atorvastatin, 80 mg, vs. pravastatin, 40 mg	Composite endpoint of death, myocardial infarction, revascularization, or stroke	0.84 (0.70–0.99)	Atorvastatin associated with a lower composite endpoint
A to Z [30]	2004	4499 patients with ACS	Simvastatin, 40 mg for 1 month, 80 mg thereafter, vs. placebo for 4 months, then simvastatin, 20 mg	Death from CV causes, non-fatal MI, or resuscitated cardiac arrest.	0.89 (0.78–1.01)	No difference in primary endpoint; 16% reduction in the composite endpoint of death from any cause, non-fatal MI, readmission for ACS, or stroke.
IMPROVE-IT [36]	2015	18,144 patients with ACS	Simvastatin plus ezetimibe vs. Simvstatin	Composite endpoint of CV death, major coronary event, or non-fatal stroke	0.94 (0.89–0.99)	Simvastatin plus ezetimibe reduced the composite endpoint

**Table 3 metabolites-13-00916-t003:** Main clinical studies that have assessed the PCSK9 inhibitors in the post-ACS setting.

Study	Year	Study Design (n. of Included Patients)	Investigational Drug (Therapeutic Regimen)	Population Characteristics	Main Results	Safety Outcomes
ODYSSEY OUTCOMES [90]	2018	RCT (18,924)	Alirocumab (75 mg every 2 weeks) §	Patients with a previous ACS event (1–12 months earlier) and LDL-C > 70 mg/dL. * and receiving statin with high efficacy at the maximum tolerated dose.	MACE (including death from CHD, nonfatal MI, fatal or nonfatal ischemic stroke, or unstable angina requiring hospitalization) incidence reduction (HR 0.85; 95% CI, 0.78 to 0.93; *p* < 0.001).	AE incidence was similar in the alirocumab group and the placebo group (3.8% vs. 2.1%, *p* < 0.001)
FOURIER [111]	2020	RCT prespecified secondary analysis (27,564)	Evolocumab (either 140 mg every 2 weeks or 420 mg monthly)	Patients with ACVD treated with statins	In patients with a recent MI (4 weeks–12 months prior randomization), as compared to placebo, evolocumab reduced the risk of MACE (CV death, AMI, stroke, hospitalization for unstable angina, or coronary revascularization) (HR, 0.81; 95% CI, 0.70–0.93).	No significant difference between the study groups with regard to AE, except for injection-site reactions that were more frequent in evolocumab treatment group.
EVOPACS [33]	2019	RCT (308)	Evolocumab 420 mg (at randomization and after 4 weeks)	Patients hospitalized for ACS with high LDL-C levels **	After 8 weeks, between treatment groups, difference in mean percentage change from baseline was −40.7% (95% CI: −45.2 to −36.2; *p* < 0.001)	Similar rates of any AE and serious AE between treatments groups.
EPIC-STEMI [34]	2022	RCT (68)	Alirocumab 150 mg (prior PCI, and at 2 and 4 weeks)	Patients with STEMI undergoing PCI	–22.3% between-group difference in LDL-C reduction (*p* < 0.001)	
VCU-AlirocRT [35]	2018	RCT (20)	Alirocumab 150 mg (within 24 h of presentation).	Patients with NSTEMI and LDL-C > 70 mg/dL despite high intensity statin therapy	Alirocumab reduced LDL-C from baseline to 14 days by 64 mg/dL (−96 to −47) compared with placebo +1 mg/dL (−25 to +16)]	No between-group difference in serious AE occurrence attributable to the study treatment.

§ Alirocumab was adjusted to achieve LDL-C levels of 25–50 mg. * Patients had LDL-C > 70 mg/dL, non-high-density lipoprotein cholesterol > 100 mg/dL, or an apolipoprotein B > 80 mg/dL. ** (≥70 mg/dL on high-intensity statin for ≥4 weeks; ≥89 mg/dL on low- or moderate-intensity statin; or ≥124 mmol/L on no stable dose of statin). Abbr: ACS: acute coronary syndrome; AE adverse event; CHD, coronary heart disease; CI, confidence interval; HR, hazard ratio; MI, myocardial infarction; LDL-C, low-density lipoprotein cholesterol; PCI, percutaneous coronary intervention; PCSK9, proprotein convertase subtilisin/kexin type 9; NSTEMI, non-ST-segment elevation myocardial infarction; STEMI, ST-segment elevation myocardial infarction.

**Table 4 metabolites-13-00916-t004:** International Guidelines Overview.

ESC/EAS 2019 Guidelines for the Management of Dyslipidaemias [25]	ESC/EAS 2020 Guidelines for the Management of NSTE-ACS [115]	ESC/EAS 2021 Guidelines on Cardiovascular Disease Prevention in Clinical Practice [26]	2018 AHA/ACC Guidelines on the Management of Blood Cholesterol [114]
In secondary prevention for patients at very high risk, an LDL-C reduction of ≥50% from baselined and an LDL-C goal of <1.4 mmol/L (<55 mg/dL) are recommended **Class A level A**	Statins are recommended in all NSTE-ACS patients. The aim is to reduce LDL-C by ≥50% from baseline and to achieve LDL-C < 1.4 mmol/L (<55 mg/dL) **Class A level A**	In patients with established ASCVD, LLT with an LDL-C goal of <1.4 mmol/L (55 mg/dL) and a >50% reduction in LDL-C vs. baseline is recommended **Class A level A**	In patients who are 75 years of age or younger with clinical ASCVD, high-intensity statin therapy should be initiated or continued with the aim of achieving a 50% or greater reduction in LDL-C levels **Class A Level A**
For patients with ASCVD who experience a second vascular event within 2 years (not necessarily of the same type as the first event) while receiving maximally tolerated statin-based therapy, an LDL-C goal of <1.0 mmol/L (<40 mg/dL) may be considered **Class IIb Level B**	If the current NSTE-ACS episode is a recurrence within less than 2 years of a first ACS while receiving maximally tolerated statin-based therapy, an LDL-C goal of <1.0 mmol/L (<40 mg/dL) may be considered **Class IIb Level B**	**The target isn’t specified**	
If the goals are not achieved with the maximum tolerated dose of a statin, combination with ezetimibe is recommended **Class I Level B (the time of re-evaluation is 4–6weeks)**	If the LDL-C goal is not achieved after 4–6 weeks with the maximally tolerated statin dose, combination with ezetimibe is recommended **Class I Level B**	If the goals are not achieved with the maximum tolerated dose of a statin, combination with ezetimibe is recommended **Class I Level B (the time of re-evaluation is not clearly announced)**	In patients with clinical ASCVD who are on maximally tolerated statin therapy and are judged to be at very high risk and have an LDL-C level of 70 mg/dL or higher (1.8 mmol/L), it is reasonable to add ezetimibe therapy **Class IIb Level B-R**
For secondary prevention in patients at very high risk and not achieving their goal on a maximum tolerated dose of a statin and ezetimibe, a combination with a PCSK9 inhibitor is recommended **Class I Level A**	If the LDL-C goal is not achieved after 4–6 weeks despite maximally tolerated statin therapy and ezetimibe, the addition of a PCSK9 inhibitor is recommended **Class I Level B**	For secondary prevention in patients not achieving their goals on a maximum tolerated dose of a statin and ezetimibe, combination therapy including a PCSK9 inhibitor is recommended **Class I Level A**	In patients with clinical ASCVD who are judged to be at very high risk and considered for PCSK9 inhibitor therapy, maximally tolerated LDL-C lowering therapy should include maximally tolerated statin therapy and ezetimibe **Class I Level B-NR**
If a statin-based regimen is not tolerated at any dosage (even after rechallenge), ezetimibe should be considered **Class IIa Level C**			
If a statin-based regimen is not tolerated at any dosage (even after rechallenge), a PCSK9 inhibitor added to ezetimibe may also be considered **Class IIb Level C**			
Treatment with statins is recommended for older people with ASCVD in the same way as for younger patients **Class I Level A**			In patients older than 75 years of age with clinical ASCVD, it is reasonable to initiate moderate or high-intensity statin therapy after evaluation of the potential for ASCVD risk reduction, adverse effects, and drug–drug interactions, as well as patient frailty and patient preferences– **Class IIa BR**

Abbr: LDL-C: low-density lipoprotein cholesterol; NSTE-ACS: non-ST-segment elevation myocardial infarction; ASCVD: atherosclerotic cardiovascular disease; LLT: lipid-lowering treatment; ACS acute coronary syndrome; PCSK9 proprotein convertase subtilisin/kexin type 9.

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
