# Peer review of "Appropriateness of Dyslipidemia Management Strategies in Post-Acute Coronary Syndrome: A 2023 Update"

_metabolites, 2023, doi:10.3390/metabo13080916_

Round 1
Reviewer 1 Report
In this review, Fabiana Lucà et al aimed to discuss the possible lipid-lowering strategies in post-ACS and to identify those patients who might benefit most from more powerful treatments and up-to-date management.
The work is interesting and very well written, providing data from RCT and real-world. However, there are some minor comments to address.
Minor comments.
1) There has been a shift in affiliations, they do not match and maybe one is missing.
2) There is some missing punctuation as well as some spacing or typing errors (VC instead of CV or target instead of targhet).
3) When you cited several clinical trials on page 4, lines 180-183, references are required.
None
Author Response
REVIEWER 1
In this review, Fabiana Lucà et al aimed to discuss the possible lipid-lowering strategies in post-ACS and to identify those patients who might benefit most from more powerful treatments and up-to-date management.
The work is interesting and very well written, providing data from RCT and real-world. However, there are some minor comments to address.
Minor comments.
1) There has been a shift in affiliations, they do not match and maybe one is missing.
2) There is some missing punctuation as well as some spacing or typing errors (VC instead of CV or target instead of targhet).
3) When you cited several clinical trials on page 4, lines 180-183, references are required.
- ANSWER: We thank the reviewer for these kind comments. According to the reviewer’s request, references to the trial citations have been added (see page 4, lines 180-180). Moreover, we corrected affiliations, punctuation and typing errors
Reviewer 2 Report
This is a well written and comprehensive review/resume of the state-of-art therapeutic possibilities and advances in the (preventive) treatment of dyslipidemia of CVD/ACS.
As is I have no further comments.
Author Response
REVIEWER 2
- Suggestions for Authors
This is a well written and comprehensive review/resume of the state-of-art therapeutic possibilities and advances in the (preventive) treatment of dyslipidemia of CVD/ACS.
As is I have no further comments.
ANSWER: We thank the reviewer for these kind comments.
Reviewer 3 Report
This is a good review of lipid-lowering therapy for patients with acute coronary syndromes and is analyzed in detail, including recent data. Although I consider it a good and educational review article, I would like to request the following revisions.
#1 Many paragraphs are changed in one or two sentences, often in the first half of the review, making it very difficult to read. The authors should group several documents together as paragraphs to make sense.
#2 The final numbering is up to 15. There are so many that individual drugs should be indicated by a subtitle, such as 2-a) statin.
I don't see any problem with English.
Author Response
REVIEWER 3
This is a good review of lipid-lowering therapy for patients with acute coronary syndromes and is analyzed in detail, including recent data. Although I consider it a good and educational review article, I would like to request the following revisions.
- Many paragraphs are changed in one or two sentences, often in the first half of the review, making it very difficult to read. The authors should group several documents together as paragraphs to make sense.
- The final numbering is up to 15. There are so many that individual drugs should be indicated by a subtitle, such as 2-a) statin.
ANSWER: We agree with the reviewer We modified the paper accordingly. After the revision, the total number of paragraphs is nine. We unified paragraph 2 with the five following (2a, 2b, 2 c, 2 d, 2e). Moreover, we unified paragraphs 12 and 13 in paragraph 7, see page 14-15, lines 488-514